# Multi-Object Tracking with Confidence-Based Trajectory Prediction Scheme

**DOI:** 10.3390/s25237221

**Published:** 2025-11-26

**Authors:** Kai Yi, Jiarong Li, Yi Zhang

**Affiliations:** 1Intelligent Policing Key Laboratory of Sichuan Province, Luzhou 646000, China; yikai@scpolicec.edu.cn; 2College of Computer Science, Sichuan University, Chengdu 610065, China; ljr@stu.scu.edu.cn

**Keywords:** multi-object tracking, confidence score, trajectory prediction, data association

## Abstract

Multi-Object Tracking (MOT) aims to associate multiple objects across consecutive video sequences and maintain continuous and stable trajectories. Currently, much attention has been paid to data association problems, where many methods filter detection boxes for object matching based on the confidence scores (CS) of the detectors without fully utilizing the detection results. Kalman filter (KF) is a traditional means for sequential frame processing, which has been widely adopted in MOT. It matches and updates a predicted trajectory with a detection box in video. However, under crowded scenes, the noise will create low-confidence detection boxes, causing identity switch (IDS) and tracking failure. In this paper, we thoroughly investigate the limitations of existing trajectory prediction schemes in MOT and prove that KF can still achieve competitive results in video sequence processing if proper care is taken to handle the noise. We propose a confidence-based trajectory prediction scheme (dubbed ConfMOT) based on KF. The CS of the detection results is used to adjust the noise during updating KF and to predict the trajectories of the tracked objects in videos. While a cost matrix (CM) is constructed to measure the cost of successful matching of unreliable objects. Meanwhile, each trajectory is labeled with a unique CS, while the lost trajectories that have not been updated for a long time will be removed. Our tracker is simple yet efficient. Extensive experiments have been conducted on mainstream datasets, where our tracker has exhibited superior performance to other advanced competitors.

## 1. Introduction

MOT has been highly spotlighted recently; it deals with the detection of multiple objects across video frames and identifier assignment for each trajectory. It plays a vital role in intelligent video surveillance, military surveillance, autonomous driving, etc. Over the past few years, MOT has benefited greatly from the rapid development of advanced object detection schemes. Primarily, the tracking-by-detection (TBD) paradigm has achieved tremendous success in MOT. Most existing works divide MOT into detection and association tasks and solve them independently under the TBD framework [1,2]. Typically, they first apply object detectors to detect known targets and infer the trajectories via different matching strategies for cross-frame data association. For instance, MOTRv2 [3] integrates YOLOX with the MOTR tracker, where YOLOX creates high-quality object proposals to facilitate the association process by MOTR. However, TBD has a notable problem: the tracking performance is highly reliant on the detection results (i.e., the core impact on the tracking performance is the threshold of the detector and subsequent matching mechanisms). Many existing works only retain detection results above the threshold for association and tracking and discard detection results below this threshold directly. But a fixed threshold of the detector has difficulty coping with complex situations where multiple objects interact. Simply discarding those objects will cause a large number of missed detections, which are irreversible, reducing overall tracking performance. Classical TBD-based methods include SORT [4], DeepSORT [1], and ByteTrack [2]. Among them, SORT [4] simply uses intersection over union (IoU) and motion information to measure matching similarity, and it creates new trajectories and rejects lost ones. DeepSORT [1] employs a matching cascade and calculates Mahalanobis distance between predicted track boxes and detection boxes (instead of IoU), since the accuracy of the inactive tracks decreases over time. However, the Mahalanobis distance is only used to avoid irrational assignments, which is not suitable for matching. ByteTrack [2] associates the low-confidence detections that are unmatched in the first stage, which indeed pushes the record of MOT to a new level and therefore has been adopted by many works. However, high-confidence detections are usually preferred over low-confidence ones, while low-confidence detections will no longer be assigned to inactive trajectories. Fortunately, with the continuous development of object detection, the detection results are becoming increasingly reliable. Under such cases, we deal with the detections with different CS in a separate way. We match the high-confidence trajectories in the first stage and remedy the potential trajectories with low CS in the second stage.

Apart from detection, frequent object occlusions under crowded scenes are still the major challenge for MOT [5], causing IDS. Various appearance models with effective feature learning methods are employed to calculate feature similarity so as to correct object ID when occlusions occur [6]. TransCenter [7] advocates the use of image-related dense detection queries and efficient sparse tracking queries under their query learning networks (QLNs). AMtrack [8] is built upon an encoder–decoder Transformer architecture, which realizes data association across frames via evolving a set of track predictions. However, it has a slower inference speed, and its predictive ability on linear motion is not as good as that of KF. DeNoising-MOT [9] learns the denoising process in an encoder–decoder structure and develops a cascaded mask strategy to prevent mutual suppression between neighboring trajectories. However, as the scenes become more complex, the tracking performances of the above methods will still be severely affected by occlusion. Although recent improvements in detector performance are conducive to correct object associations, even the best detectors make false predictions. There remains room for improvement in existing methods using detections.

The linear KF algorithm has long been widely used in visual object tracking [2,10]. During the update process, the measurement noise is set to a constant value. For a reliable detection result, the measurement noise should be low accordingly. To obtain more accurate update results, we adjust the measurement noise based on the CS (output by the detector) so as to ensure the quality of the tracking results. During the matching process between the prediction and detection boxes, IoU and feature similarities are usually used to form CM (to measure the discrepancy between the predicted target position and the detection box) of the Hungarian algorithm. When computing CM, the reliability of the current bounding box is often ignored. We therefore fuse the CS into CM to improve its reliability. In some cases, some trajectories exist for only a period of time but discontinue in the following frames, which disturbs the subsequent trajectory matching and causes incorrect association. We regard them as lost and hence delete them in due course. Through analyzing the connection between the detection and associations, we believe the detection results have not been fully utilized. Our results show that the performance of the current TBD-based methods can be further improved by combining the CS of the detection results. Therefore, an appropriate detection processing scheme from a tracking perspective is required to bridge the gap between detection and tracking applications.

In a nutshell, we have made contributions in three key stages of MOT, including data association, trajectory prediction, and trajectory management:1.In the data association stage, confidence scores are integrated into the cost matrix to improve matching accuracy and achieve more reliable assignment results.2.In the trajectory prediction stage, a confidence-based trajectory prediction scheme has been proposed based on KF, which achieves more accurate prediction performance by controlling the measurement noise in KF.3.In the trajectory management stage, a trajectory deletion scheme has been proposed to determine the duration of trajectories and delete less reliable trajectories to avoid possible incorrect matches.

The experimental result shows that our tracking scheme further improves the performance of the current TBD-based methods. On MOT Challenge benchmarks, ConfMOT ranks among the top on MOT17 [11] and MOT20 [12]. ConfMOT achieves 64.5% HOTA and 80.5% MOTA on the MOT17 dataset, and 62.9% HOTA and 78.3% MOTA on the MOT20.

The rest of the paper is organized as follows: Section 2 briefly reviews related work, Section 3 describes our proposed method in detail. Experimental results are provided in Section 4 with ablation studies. Finally, a conclusion is drawn in Section 5.

## 2. Related Works

With the rapid development of object detection [13], the current research hotspots of visual tracking [2,14] have been shifted from designing complex and individual trackers to subsequent data association schemes that are built upon advanced detectors. The main goal is to maintain the correct trajectories of the objects.

### 2.1. Motion Models

Most of the recent TBD-based methods are based on motion models (under the assumption of constant velocity). As one of the classical models, KF is a Bayes filter that predicts and updates in a recursive way. The ground-truth state is an unobserved Markov process, while the measurement is observed from a hidden Markov model. The measurement noise is the noise that exists in the signal observation process. OC-SORT [14] uses virtual trajectories to smooth parameters for solving the cumulative error of KF. ByteTrack [2], on the other hand, uses a normal linear KF to associate every detection box to recover true objects and filter out the background elements. However, the above two methods apply the same KF to all targets, achieving trajectory prediction quickly and efficiently, but ignoring the differences between various detections. You et al. [15] propose a novel Spatial–Temporal Topology-based Detector (STTD) algorithm and introduce a topology structure for the dynamics of moving targets. It indeed reduces false positives, but since it only considers group motion between targets to build topology, it hence overlooks camera motion. Li et al. [6] present a TBD-based framework for MOT, where a deep association network is developed (followed by detection) to infer the association degree of objects. Despite some advantages in correlation operation, the object detection and tracking tasks are somewhat independent of each other, where two deep feature extractions are needed in each stage. GIAOTracker [16] proposes a modified KF with confidence scores. Khurana et al. [17] adopt depth estimates from a monocular depth estimator to forecast the trajectories of occluded people. However, they rely heavily on appearance features, which introduces a high computational cost. Wang et al. [18] published a novel approach to tackle long-term vehicle tracking without appearance information. But it cannot be extended to other types of objects. Tracktor++ [19] employs camera motion compensation for frame alignment. The former method performs global association by developing an appearance-free link model to address missing detection and missing association problems. While the latter employs global information and optimization strategies to associate object trajectories. But it is not applicable to online real-time tracking.

### 2.2. Cost Matrix

In order to update the objects’ trajectories, the prediction boxes obtained from KF prediction phase need to be matched with the detection boxes obtained from the detector. OC-SORT [14] claims that using IoU alone is sufficient to generate good results if the detector is accurate enough. But it ignores the impact of detection differences on the cost matrix. MOTDT [20] constructs the cost matrix of the tracking processes with appearance features and IoU respectively. FairMOT [21], on the other hand, fuses IoU and appearance feature, but it does not consider the negative impact of an unreliable cost matrix. The above 2 methods combine multiple indicators to form a cost matrix, which can solve the confused assignment problem to some extent. However, the extraction of appearance information also slows down the tracking speed in return.

### 2.3. Trajectories Management

Most trackers set different durations for lost trajectories for different datasets. For example, JDE [10] and FairMOT [21] save the unmatched trajectories for 30 frames, while ByteTrack [2] keeps lost trajectories for 14 frames for MOT17-06 and 25 frames for MOT17-14 according to the length of the videos. However, these parameter settings require pre-processing of the video, since the time of disappearing trajectory varies due to different occlusion situations. It would be impractical to make different settings manually for different datasets. From a practical perspective, a unified trajectory processing scheme should be developed that is applicable to different datasets, while different time intervals should be assigned to different kinds of lost trajectories.

## 3. Method

The block diagram of our tracker is drawn in Figure 1. Basically, the CS is incorporated into the tracking process, where the detection boxes with high and low scores are taken care of by the first and the second association processes, respectively, to ensure correct associations of trajectories.

### 3.1. Confidence-Based Adaptive KF

KF (with a constant velocity) is commonly adopted in object tracking tasks in building a motion model. It follows the Markov assumption that the current system state is only related to the state of its previous moment, which is irrelevant to other earlier states. Then the previous state vector and the state observed at the current frame are needed to estimate the current state. Specifically, at time *t*, the hidden state of an object $X_{t}$ is written as(1)$$\begin{array}{c} \begin{array}{c} \;x_{t}={\left[u_{t},v_{t},a_{t},h_{t},{\dot{u}}_{t},{\dot{v}}_{t},{\dot{a}}_{t},{\dot{h}}_{t}\right]}^{T}, \end{array} \end{array}$$
where ($u_{t}$, $v_{t}$) denotes the center position of the bounding box, $h_{t}$ denotes the height of the detection box, $a_{t}$ is the aspect ratio of the width and the height. ${\dot{u}}_{t},{\dot{v}}_{t},{\dot{a}}_{t},{\dot{h}}_{t}$ represent their corresponding derivatives.

KF consists of a prediction step and an update step. In the prediction step, KF generates the estimated state variables (along with their uncertainties), which will then be updated with a weighted average of the estimated state and measurement. Since there is no active control in multi-object tracking, the state vector at frame *t* is represented by(2)$$\begin{array}{c} {\hat{x}}_{t|t-1}=F_{t}x_{t-1|t-1},\;P_{t|t-1}=F_{t}P_{t-1|t-1}F_{t}^{T}+Q_{t}, \end{array}$$

$P_{t|t-1}$ is the covariance matrix of the posterior state. $F_{t}$ is the state transition matrix and $Q_{t}$ is the covariance of noise, which are calculated as(3)$$\begin{array}{cc} F_{t} & =\left[\begin{array}{cc} I_{4\times 4} & I_{4\times 4}\\ O_{4\times 4} & I_{4\times 4} \end{array}\right],\\ Q_{t} & =\mathrm{diag}({\left(\delta_{p}w_{t-1|t-1}\right)}^{2},{\left(\delta_{p}h_{t-1|t-1}\right)}^{2},\\ \; & \;{\left(\delta_{p}w_{t-1|t-1}\right)}^{2},{\left(\delta_{p}h_{t-1|t-1}\right)}^{2},\\ \; & \;{\left(\delta_{v}w_{t-1|t-1}\right)}^{2},{\left(\delta_{v}h_{t-1|t-1}\right)}^{2},\\ \; & \;{\left(\delta_{v}w_{t-1|t-1}\right)}^{2},{\left(\delta_{v}h_{t-1|t-1}\right)}^{2}) \end{array}$$

$\delta_{v}$ and $\delta_{p}$ are the parameters of the noise. Meanwhile, the observation state vector (i.e., measurement by the detector) is expressed by(4)$$\begin{array}{c} z_{t}={\left[z_{x},z_{y},z_{w},z_{h}\right]}^{T}=Hx_{t}+r_{t},\;H=\left[I_{4\times 4}\;O_{4\times 4}\right], \end{array}$$

Here $r_{t}$ is the measurement noise. *H* is the observation matrix. For classical KF, the update process is implemented by computing the Kalman gain as(5)$$\begin{array}{c} \begin{array}{cc} \; & K_{t}=P_{t|t-1}H^{T}{\left(HP_{t|t-1}H^{T}+R_{t}\right)}^{-1}, \end{array} \end{array}$$

Then the current object state and covariance matrix of the posterior state are updated as(6)$$\begin{array}{c} \begin{array}{cc} \; & x_{t|t}={\hat{x}}_{t|t-1}+K_{t}\left(z_{t}-H{\hat{x}}_{t|t-1}\right),\\ \; & P_{t|t}=\left(I-K_{t}H\right)P_{t|t-1}, \end{array} \end{array}$$

For existing methods (e.g., SORT [4], ByteTrack [2] and OC-SORT [14]), the covariance of the measurement noise $R_{t}$ is a fixed value for all detections. However, detection reliability varies significantly: low-confidence detections frequently contain larger localization errors. Assigning the same $R_{t}$ to all detections forces KF to over-trust unreliable observations, resulting in trajectory drift and ID switches. To mitigate this issue, we introduce an adaptive scheme based on the confidence score. The update process of KF is expressed as(7)$$\begin{array}{c} \begin{array}{cc} \; & R_{t}=\left(1-c_{t}\right)R, \end{array} \end{array}$$
where $c_{t}$∈ [0, 1] is the confidence score of the matched detection and *R* is the base covariance of the measurement noise. Therefore, Equation (Equation 5) will be modified into(8)$$\begin{array}{c} \begin{array}{cc} \; & K_{t}=P_{t|t-1}H^{T}{\left(HP_{t|t-1}H^{T}+\left(1-c_{t}\right)R\right)}^{-1}, \end{array} \end{array}$$

As can be seen, a high-confidence detection ($c_{t}$ → 1) makes a lower $R_{t}$ and higher Kalman gain, while the update process relies heavily on the measurement. Conversely, a low-confidence detection ($c_{t}$ → 0) makes a higher $R_{t}$ but a lower Kalman gain, while the update process will be affected more by the predicted state, preventing unreliable detections from destroying the trajectories.

Our proposed adaptive mechanism enables KF to dynamically balance prediction and measurement based on the confidence scores of detection, thereby improving robustness against noisy detections and reducing ID switches in crowded or occluded environments.

### 3.2. Data Association

The posterior state of frame *t* $\left({\hat{x}}_{t|t-1}\right)$ is propagated from frame t − 1 through KF. The predicted bounding boxes are then used to compute IoU with all detections at frame *t* to form CM. Then the trajectories will be associated with the detection boxes via the Hungarian algorithm based on the CM and will be updated via KF.

The detection boxes with low confidence scores are regarded as unreliable, which may contain large localization errors and have a higher cost for successful matching. If they contribute equally to CM, they may incorrectly alter trajectory estimates. To alleviate this issue, ConfMOT rectifies IoU-based costs by using the CS of each detection as follows:(9)$$\begin{array}{c} C_{i,j}=1-\frac{1+S_{j}}{2}\times \left(1-c_{i,j}^{iou}\right), \end{array}$$
where $S_{j}$ is the CS of the *j*th detection and $c_{i,j}^{iou}$ is the IoU between the *i*th predicted box and the *j*th detection. The above equation assigns a higher cost to unreliable detections (low $S_{j}$) and therefore lowers their chance of being selected during the matching process and encourages high-confidence detections to dominate the association process.

### 3.3. Deletion Strategy

In previous methods, the lost trajectories were usually kept for a certain number of frames, which may be recovered if matches were found or otherwise deleted. However, the trajectories with lower CSs are essentially less reliable. We therefore create track score for each trajectory by summing up the CsS of an object on its last tracklet. For a tracklet that lasts for *n* frames, its track score is expressed as(10)$$\begin{array}{c} S_{track}=\sum_{i=1}^{n}S_{i}, \end{array}$$
where $S_{i}$ represents the CS of an object in the *i*th frame of the tracklet. The lower the track score, the less reliable the trajectory will be. Such a trajectory will be assigned a shorter time of duration before it is regarded as lost. For the lost trajectories, we use the following equation to determine the time to delete them:(11)$$S_{\mathrm{del}}=\min\;\left\{\alpha S_{\mathrm{track}}+S_{\mathrm{last}},\;1\right\}-\log\;\left(1+\beta T_{\mathrm{umatch}}\right),$$
where $S_{last}$ is the CS of the last frame of the tracklet and $T_{umatch}$ is the number of frames when the trajectory does not find a match in the following frames. α and β are two parameters.

The design of Equation (Equation 11) is quite intuitive: the first term min $\left\{\alpha S_{\mathrm{track}}+S_{\mathrm{last}},\;1\right\}$ depicts the reliability of a trajectory. $\alpha S_{\mathrm{track}}+S_{\mathrm{last}}$ aggregates both the accumulated CS along the tracklet and the CS of the last observation. It may become a large number due to a possibly large $S_{\mathrm{track}}$; in that situation, the trajectory will not be able to be deleted. We therefore apply both α and min( , 1) to restrict it to [0, 1]. The addition of $S_{\mathrm{last}}$ prevents newly generated but highly reliable trajectories from being penalized only because their durations are short. The second term $\log\;\left(1+\beta T_{\mathrm{umatch}}\right)$ imposes a time-dependent penalty on lost trajectories. It is monotonically increasing with $T_{\mathrm{umatch}}$. Since log operation increases slowly at the beginning and becomes more aggressively when $T_{\mathrm{umatch}}$ becomes large. It allows the short-term occluded objects but deletes long unmatched trajectories. The parameters α and β realize the trade-off between the contribution of historical confidence and the speed of temporal decay.

Overall, $S_{\mathrm{del}}$ is an interpretable confidence-like score. It assigns longer survival time to reliable trajectories or those that are only shortly unmatched, which shortens the unreliable trajectories or those that remain unmatched for a long time.

### 3.4. Tracking Procedure

The pseudo-code of tracking is shown in Algorithm 1, and the corresponding diagram has been drawn in Figure 2 to better understand the process. Firstly, we adopt YOLOX as our detector to obtain the bounding boxes and the corresponding CS, where the detection boxes are divided into two parts: $D_{high}$ (yellow in Figure 2) and $D_{low}$ (green in Figure 2) based on confidence thresholds $\tau_{high}$ and $\tau_{low}$(which are set to 0.6 and 0.1, respectively, according to OC-SORT [14]). The bad detection boxes with CS lower than $\tau_{low}$ are removed. Then, we use KF to predict the new locations of each trajectory in $T_{all}$ (labeled 1, 2, and 3 in Figure 2).

The first association is performed between the high score boxes $D_{high}$ and all trajectories $T_{all}$. We only use the IoU CM via Equation (Equation 9) and utilize the Hungarian algorithm to assign the detection boxes to corresponding trajectories. In particular, if the IoU cost between the detection box and the prediction box is greater than a threshold, then the matching will be rejected. The unmatched detections in the first association are kept in $D_{remain}$, and the unmatched trajectories (e.g., No. 3 in Figure 2) are kept in $T_{remain}$ (lines 14 to 18 in Algorithm 1), which will be matched with the low score boxes in the second stage association.

In the second association, $T_{remain}$ will be associated with the low score boxes $D_{low}$. The unmatched detection boxes in $D_{low}$ are regarded as the background, and unmatched trajectories are marked as lost and kept in $T_{lost}$ (lines 19 to 21 in Algorithm 1), which will be deleted after a certain period of time.

The successfully matched trajectories are updated by KF through Equation (Equation 8). For trajectories in $T_{lost}$, their CS ($S_{del}$) will be calculated via Equation (Equation 11). If they are lower than $\tau_{del}$, they will be moved from $T_{lost}$ to $T_{del}$. If a lost trajectory is recovered, it will be moved from $T_{lost}$ to T, and the track score will be recalculated.
**Algorithm 1:** Pseudo-code of tracking
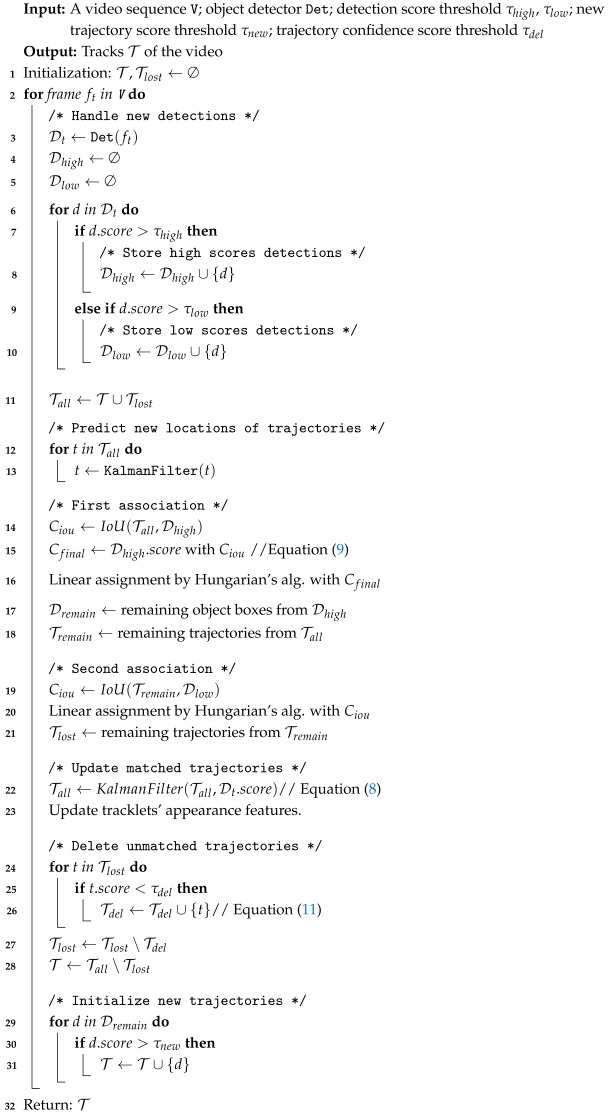


Finally, for detection boxes in $D_{remain}$, we initialize new trajectories if their CSs are higher than $\tau_{new}$ and exist for two consecutive frames (lines 29 to 31 in Algorithm 1).

## 4. Experiment

### 4.1. Datasets

The training process of MOT17 [11] is conducted jointly on CrowHuman [22], CityPersons [23], ETHZ [24], and the training set of MOT17. And we use CrowdHuman and the training set of MOT20 [12] to train MOT20, while testing is carried out on MOT17 and MOT20 under the “private detection” protocol. For the ablation study, we combine CrowdHuman and half of the training set of MOT17 as the training set and test our tracker on the second half of MOT17. The initial parameters of our model are pre-trained on COCO [25].

### 4.2. Metrics

We adopt Higher-Order Tracking Accuracy (HOTA), Multi-Object Tracking Accuracy (MOTA), ID F1 Score (IDF1) (the ratio of correctly identified detections over the average number of ground-truth and computed detections), and Number of Identity Switches (IDS) as the main metrics to evaluate the performance of our model. Considering MOTA is mainly used to assess detection performance while IDF1 characterizes the ability to keep constant identities, we use HOTA as the final evaluation metric for ranking, which is a more comprehensive indicator to reflect the general performance of Detection Accuracy, association, and localization. In addition, we adopt Association Accuracy (AssA), Association Precision (AssPr), Association Recall (AssRe), Detection Accuracy (DetA), and Localization Accuracy (LocA) to further compare the trackers with similar performances. AssA, AssPr, and AssRe are used to measure association performance, while DetA is used for detection quality.

### 4.3. Implementation Details

All the experiments are implemented in PyTorch 1.13.1 and under NVIDIA Tesla V100 GPU. We use YOLOX as the detector with YOLOX as the backbone. The datasets and training strategy are described in Section 4.1. The optimizer is SGD with a weight decay of $5\times 10^{-4}$ and a momentum of 0.9, and the initial learning rate is set to $10^{-3}$. The parameters of the tracker are the same as the baseline. The high- and low-confidence thresholds $\tau_{high}$ and $\tau_{low}$ are empirically set to 0.6 and 0.1, respectively. If the CSs of unmatched detections are higher than 0.7 [2], we initialize new trajectories starting from them. The lost trajectories will be deleted if their CSs are lower than 0.1. GSI is adopted as a post-processing method.

### 4.4. Experimental Results

Extensive experiments were conducted on MOT17 [11] and MOT20 [12] to testify to the effectiveness of our tracker.

#### 4.4.1. Results on MOT17

The video sequences in MOT17 are filmed by both static and moving cameras. The comparative results on MOT17 are listed in Table 1. Generally, ByteTrack [2], OC-SORT [14], SCTrack [26], etc., and ConfMOT outperform other trackers with some margins. Among them, ConfMOT surpasses ByteTrack by 1.4% and 2.0% in HOTA and MOTA, respectively, with lower IDS as well. Meanwhile, we lead OC-SORT in HOTA, MOTA, and IDF1. Our obvious advantage in HOTA is attributed to the proposed early trajectory deletion strategy, which ensures the tracking accuracy.

To further investigate the subtle differences among ByteTrack [2], OC-SORT [14], and our tracker, we compare another five metrics: AssA, AssPr, AssRe, DetA and LocA. As mentioned in Section 4.2, the first three metrics reflect the data association ability of a tracker, while DetA and LocA demonstrate the accuracies of object detection and localization, respectively. As shown in Table 2, except that we lag behind OC-SORT in AssPr, ConfMOT outperforms the other two trackers in all other metrics. The reason is that OC-SORT is stronger than our trackers in association ability, but weaker in detection. It is worth noting that ConfMOT ranks first in AssRe, which reflects the accurate prediction of the target trajectories.

A group of visualization results of ConfMOT on the test set of MOT17 is shown in Figure 3. The MOT17-01 scene is a normal outdoor scenario with pedestrians walking around. Apparently, the color of the bounding boxes does not change or swap, indicating constant identities during the tracking process. MOT17-03 is a crowded outdoor scene, in which our tracker still completes the tracking task without obvious miss detections or ID switches. MOT17-06 is a normal street scene, and it is worth noting that the old man with dark blue box number 27 (in the middle of the scene) always remains unchanged after several overlaps by other pedestrians, which indicates the ability of our tracker to deal with occlusions. In MOT17-07, the video is taken by moving cameras, and there is a man sitting in the left corner in the first frame, who is moving closer in the following frames with a constant ID. MOT17-8 shows a tracking scenario with varying lighting conditions. Still, our tracker demonstrates robustness in tracking multiple targets with fixed IDS. Lastly, MOT17-14 is a video sequence containing small targets that is captured on a moving box. Clearly, we can correctly detect and locate the small targets over a moving viewpoint.

#### 4.4.2. Results on MOT20

MOT20 is a more challenging benchmark with crowded scenes. As shown in Table 1, ConfMOT ranks first in HOTA, MOTA and IDF1 due to our proposed deletion strategies, but it also causes slightly higher IDS, since the lost tracks are difficult to match. Like MOT17, we also compare AssA, AssPr, AssRe, DetA, and LocA among OC-SORT, ByteTrack, and ConfMOT. As shown in Table 3, ConfMOT surpasses both OC-SORT and ByteTrack in all five metrics, which reflects stronger association accuracy and more precise localization. The deletion of unreliable lost trajectories in the early stage ensures the overall tracking performances, while the integration of the confidence-based cost matrix guarantees the robustness of the tracker during the long period of the tracking process.

A group of visualization results of ConfMOT on the test set of MOT20 is shown in Figure 4, including indoor and outdoor scenarios under different lighting conditions. As mentioned earlier, MOT20 includes crowded scenes. MOT20-04 is an outdoor scene at night, while MOT20-06 and MOT20-08 are daytime scenarios. No matter under the scene, our tracker maintains excellent tracking performances, which is less sensitive to lighting conditions, and the ID of each target remains largely unchanged.

#### 4.4.3. Comprehensive Analysis

As mentioned earlier, ByteTrack [2], OC-SORT [14] and our tracker achieve better results than the other competitors. Here, we further analyze the advantages/disadvantages of the three methods. ByteTrack [2] utilizes pure IoU to process all the detections. In contrast, we process the detection boxes using the cost matrix, which is adjusted by the confidence score. Our scheme matches the high-quality detection boxes in a more effective way, resulting in more stable assignment results. As a result, we achieve the best overall secondary indexes (including AssA, AssPr, AssRe, DetA and LocA). Besides, we obtain a more precise object location due to the proposed modified KF, which has been verified by the highest DetA (Detection Accuracy) we have reached. In summary, we have achieved the best overall results on MOT-17 and on MOT-20.

#### 4.4.4. Visualization of Trajectories

To further demonstrate the superiority of our tracker, a comparison among ByteTrack [2], OC-SORT [14] and ConfMOT is made in Figure 5, in which the trajectory of each object of the last 50 frames is drawn with different colors. Obviously, OC-SORT (shown in Figure 5a) has many sporadic segments of trajectories, which reflect target loss under the crowded scene, while it has almost no trajectories on the left-hand side of the scene. ByteTrack (shown in Figure 5b) has more trajectories, but there are some zigzags and stagnation, indicating incorrect matching and ID switches. In comparison, as shown in Figure 5c, the trajectories of ConfMOT appear more stable, with almost identical trajectory lengths and no broken segments. In addition, the colors of the trajectories do not change within the dense areas, indicating the robustness of ConfMOT under crowded scenes.

### 4.5. Ablation Studies

In the ablation experiment, we use YOLOX as the backbone and train our model jointly on CrowdHuman and the first half of the training set of MOT17, while the second half of the training set of MOT17 is used for validation. In this section, we will firstly verify the effectiveness of the core modules of our tracker, including KF, CM+CS, and the deletion strategy (DS). The comparative results on the MOT17 validation set is shown in Table 4.

#### 4.5.1. KF Based on the Confidence Score

As shown in Table 4, when KF (with the CS) is integrated into the tracking network, we enjoy a slight increase in both HOTA and IDF1. By incorporating confidence, KF assigns weights to different detections based on their CS, making it more robust to noise and uncertain detections. During the update of KF, the high-quality detection boxes will be assigned higher weights than the prediction boxes, making the detection boxes more dependent on the detection results. Conversely, for low-quality detection results, the detection boxes will depend more on the prediction boxes, which will be assigned higher weights.

#### 4.5.2. Deletion Strategy Based on the Confidence Score

As shown in Table 4, when the proposed lost trajectory deletion strategy has been added to the tracking architecture, the HOTA and IDF1 values increase by 0.2% and 0.4%, respectively. These results illustrate the fact that when a lost trajectory has been deleted and the subsequent detected target cannot find a match, and therefore, a new trajectory will be created. Moreover, the increased IDF1 suggests that the deleted trajectories are indeed unreliable, and our deletion strategy reduces the likelihood of incorrect matches between a detected target and the wrong trajectory. Intuitively, if a trajectory remains unmatched for a certain period of time and is re-matched again, it is quite possible that it matches with a different target (instead of the previous one). Therefore, our deletion strategy becomes very necessary to avoid target drifting caused by re-matching.

#### 4.5.3. Cost Matrix Based on the Confidence Score

When calculating the matching cost between the prediction and detection boxes, such a cost will be higher for low-quality detection boxes. As shown in Table 4, after the CS is incorporated into CM, both HOTA and MOTA increase by 0.4%. This result indicates that the integration of the confidence score indeed reduces the probability of matching low-quality detection boxes and continuation of incorrect trajectories.

## 5. Conclusions

In this paper, we thoroughly analyze the limitations of current TBD-based trackers and present ConfMOT to ensure the stable continuation of trajectories among different objects. To realize the goal, we adopt YOLOX as our detector to obtain the detection results. We adopt a two-stage association strategy and predict the trajectories based on the confidence score and Kalman filtering technique. Then, a cost matrix is constructed to measure the cost of unreliable matching. Finally, a deletion strategy is proposed to determine the duration of a trajectory and delete the less reliable trajectories. Extensive experiments have been conducted on the MOT17 and MOT20 datasets, along with ablation studies to testify to the efficacy of our tracker. The results prove that our trackers have obvious advantages in the HOTA metric compared to other methods, due to the cost matrix and trajectory deletion strategy we designed. However, the performance of our tracker in IDS is mediocre (ID switch caused by occlusion), since we delete unreliable trajectories at an early stage.

In future work, we will be engaged in developing a light-weight appearance feature extraction module to improve the robustness of the model in dealing with fast appearance changes. We will also employ specific motion modeling schemes to ensure the stability and continuation of the target trajectory.

## Figures and Tables

**Figure 1 sensors-25-07221-f001:**
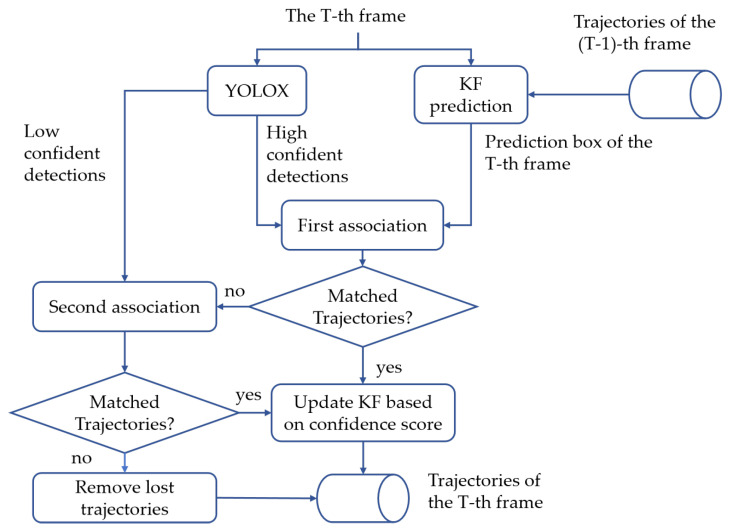
Block diagram of the proposed ConfMOT. We adopt a two-stage association scheme, where we predict and update object trajectories based on the confidence score of the detections.

**Figure 2 sensors-25-07221-f002:**
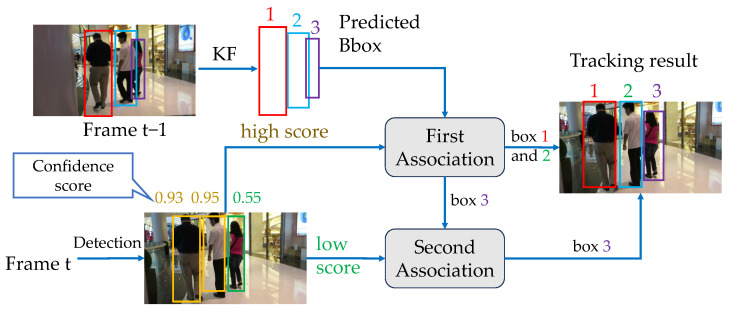
Illustration of the processing flow of the proposed method. The high-confidence detections will be processed in the first-stage association. The low-confidence ones will be sent to the second-stage association. The unmatched trajectories will be deleted after a certain period of time.

**Figure 3 sensors-25-07221-f003:**
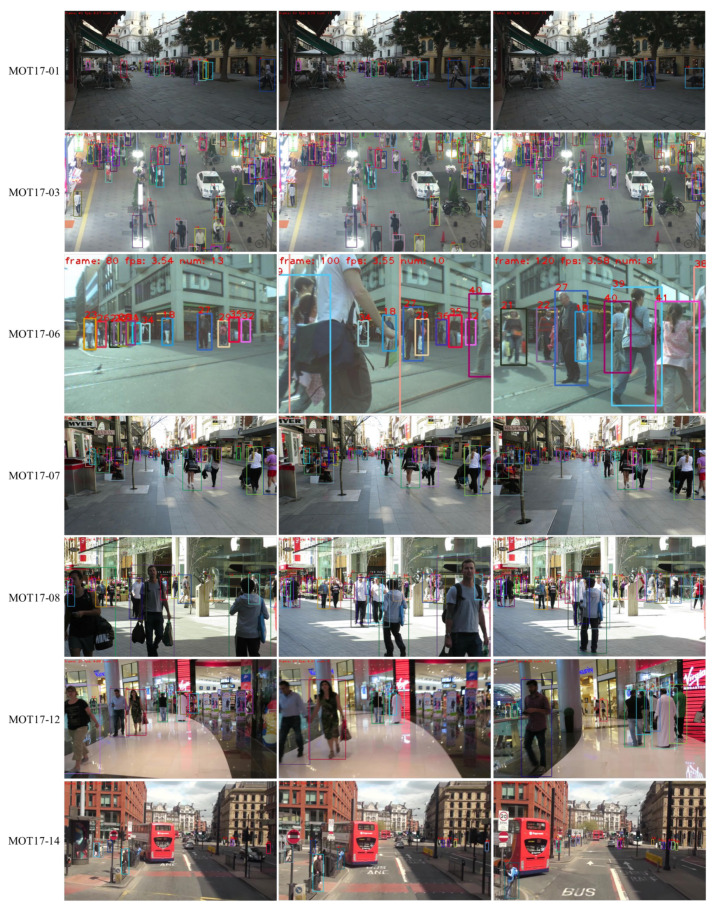
Demonstration of tracking results on the test set of MOT17; our tracker successively associates multiple objects under different scenes.

**Figure 4 sensors-25-07221-f004:**
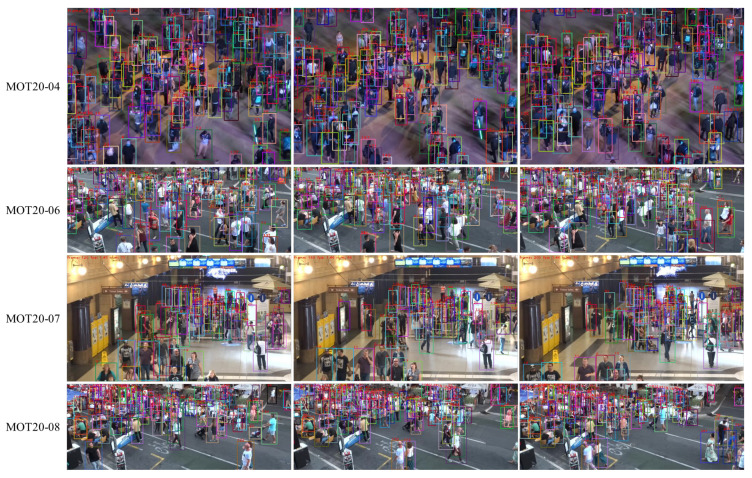
Demonstration of tracking results on the test set of MOT20; our tracker successively associates multiple objects even in crowded scenes with fewer ID switches.

**Figure 5 sensors-25-07221-f005:**
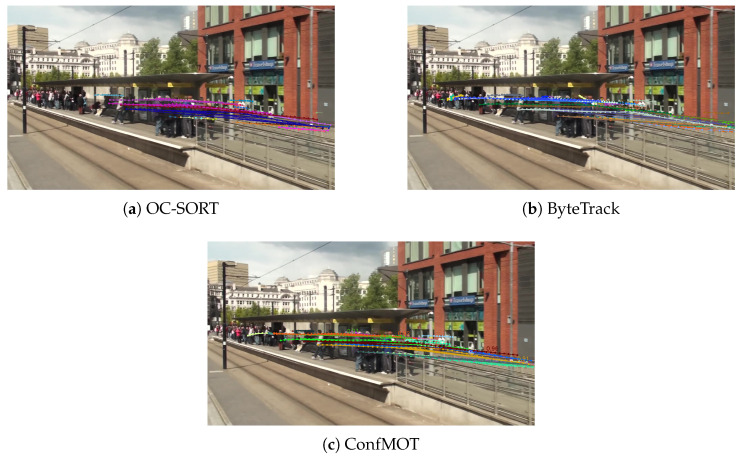
Comparison of trajectories generated by three tracking methods. Compared with ByteTrack and OC-SORT, we generate more stable trajectories with no broken segments.

**Table 1 sensors-25-07221-t001:** Evaluation on the test sets of MOT17 and MOT20. We compare our method with recent methods. The best results are shown in red and the second-best results are in blue.

Methods	MOT17					MOT20				
	**HOTA**↑	**MOTA**↑	**IDF1**↑	**IDS**↓	**FPS**↑	**HOTA**↑	**MOTA**↑	**IDF1**↑	**IDS**↓	**FPS**↑
TADN [27]	-	69.0%	60.8%	-	-	-	68.7%	61.0%	-	-
DMMTracker [28]	52.1%	67.1%	64.3%	3135	16.1	48.7%	62.5%	60.5%	2043	9.7
TransTrack [29]	54.1%	75.2%	63.5%	3603	10.0	48.5%	65.0%	59.4%	3608	7.2
TransCenter [7]	54.5%	73.2%	62.2%	4614	1.0	43.5%	61.9%	50.4%	4653	1.0
MeMOT [30]	56.9%	72.5%	69.0%	2724	-	54.1%	63.7%	66.1%	1938	-
AMtrack [8]	58.6%	74.4%	71.5%	4740	-	56.8%	73.2%	69.2%	1870	-
DNMOT [9]	58.0%	75.6%	68.1%	2529	-	58.6%	70.5%	73.2%	987	-
MeMOTR [31]	58.8%	72.8%	71.5%	-	-	-	-	-	-	-
FairMOT [21]	59.3%	73.7%	72.3%	3303	25.9	54.6%	61.8%	67.3%	5243	13.2
DiffusionTrack [32]	60.8%	77.9%	73.8%	3819	-	55.3%	72.8%	66.3%	4117	-
STDFormer-LMPH [33]	60.9%	78.4%	73.1%	5091	-	60.2%	76.2%	72.1%	5245	-
RelationTrack [34]	61.0%	73.8%	74.4%	1374	7.4	56.5%	67.2%	70.5%	4243	2.7
BGTracker [35]	61.0%	75.6%	73.8%	3735	20.7	57.5%	71.6%	71.8%	2471	12.8
ColTrack [36]	61.0%	78.8%	73.9%	1881	-	-	-	-	-	-
JDT-NAS-T1 [37]	-	74.3%	72.0%	2818	13.3	-	-	-	-	-
DcMOT [38]	61.3%	74.5%	75.2%	2682	20.4	53.8%	59.7%	67.4%	5636	10.6
MOTFR [39]	61.8%	74.4%	76.3%	2652	22.2	57.2%	69.0%	71.7%	3648	13.3
CorrTracker [40]	-	76.5%	73.6%	3369	14.8	-	65.2%	69.1%	5183	8.5
TransMOT [41]	-	76.7%	75.1%	2346	-	-	77.5%	75.2%	1615	-
MAA [42]	62.0%	79.4%	75.9%	1452	189.1	57.3%	73.9%	71.2%	1331	14.7
MOTRv2 [3]	62.0%	78.6%	75.0%	-	-	61.0%	76.2%	73.1%	-	-
PID-MOT [43]	62.1%	74.7%	76.3%	1563	19.7	57.0%	67.5%	71.3%	1015	8.7
GHOST [44]	62.8%	78.7%	77.1%	2325	-	61.2%	73.7%	75.2%	1264	-
GGSTrack [45]	62.8%	80.2%	-	1689	58.0	61.8%	75.1%	-	1498	15.3
ScoreMOT [46]	63.0%	79.8%	76.7%	4007	25.6	62.3%	77.7%	75.6%	1440	16.2
ByteTrack [2]	63.1%	80.3%	77.3%	2196	29.6	61.3%	77.8%	75.2%	1223	17.5
OC-SORT [14]	63.2%	78.0%	77.5%	1950	29.0	62.1%	75.5%	75.9%	913	18.7
AM-SORT [47]	63.3%	78.0%	77.8%	-	-	62.0%	75.5%	76.1%	-	-
SCTrack [26]	63.5%	79.4%	77.7%	2022	-	61.4%	75.6%	76.1%	837	-
ConfMOT	64.5%	80.5%	79.3%	1980	26.1	62.9%	78.3%	76.1%	1359	15.2

**Table 2 sensors-25-07221-t002:** Further comparison of different tracking methods on the MOT17 dataset. The best results are highlighted in red and the second-best results are in blue.

Tracker	AssA (%)↑	AssPr (%)↑	AssRe (%)↑	DetA (%)↑	LocA (%)↑
ByteTrack [2]	62.0	76.0	68.2	64.5	83.0
OC-SORT [14]	63.4	80.8	67.5	63.2	83.4
ConfMOT (Ours)	63.8	77.9	70.0	64.9	83.4

**Table 3 sensors-25-07221-t003:** Further comparison with ByteTrack on the testing set of MOT20. The best results are shown in **bold**.

Tracker	↑	AssPr↑	AssRe↑	DetA↑	LocA↑
ByteTrack [2]	59.6%	74.6%	66.2%	63.4%	83.6%
OC-SORT [14]	60.5%	75.1%	67.1%	64.2%	83.9%
ConfMOT	**61.4**%	**77.1**%	**67.8**%	**64.6**%	**84.7**%

**Table 4 sensors-25-07221-t004:** Ablation study on the MOT17 validation set. The best results are shown in **bold**.

KF	DS	CM+CS	HOTA↑	MOTA↑	IDF1↑
			67.8%	77.9%	79.6%
✔			68.1%	77.9%	79.8%
✔	✔		68.3%	78.0%	80.2%
✔	✔	✔	**68.7%**	**78.4**%	**80.5**%

## Data Availability

The original contributions presented in this study are included in the article. Further inquiries can be directed to the corresponding author.

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
