# Peer review of "Multi-Object Tracking with Confidence-Based Trajectory Prediction Scheme"

_sensors, 2025, doi:10.3390/s25237221_

Round 1

Reviewer 1 Report

Comments and Suggestions for Authors

The authors propose a Confidence-based Trajectory prediction scheme (ConfMOT) based on KF. The confidence score (CS) of the detection results is used to improve the accuracy of path association which is taken in two steps. The comprehensive experiments have been conducted on mainstream datasets to show the scheme having superior performance over that of other advanced competitors. Some comments are listed as follows.

  1. Taking the CS into account is a novel idea, but the reason of taking two steps in association process is not clearly explained. It seems to me that they are doing the same job (the linear Hungarian assignment). Where is your improvement comes from? What if we take three steps, even more? Could it perform better?
  2. The block diagram in figure 1 is quite confused.
  3. The variables used in equations should be described. E.g., in Eq2, c_t means what?
  4. Why the equations give the effects what you claim? E.g., Eq. 2, Eq. 5 and Eq.7 are the most important equations of your work, but unfortunately they are not well explained.
  5. The author did a good job in comparing ConfMOT with other works. But it seems to me the improvements are not very significant.

Reviewer 2 Report

Comments and Suggestions for Authors

REVIEW OF

Multi-Object Tracking with Confidence-based Trajectory prediction Scheme

BY

Kai Yi, Jiarong Li and Yi Zhang

The paper describes a new (perhaps more accurately, an improved) method for Multi-Object Tracking (MOT), i.e., the automatic following of multiple objects in video (a sequence of frames). There is a large body of work on this topic; the typical and classical approach is tracking-by-detection (first the detector finds objects in each frame, then the tracker links these detections across frames to form continuous trajectories). The authors themselves nicely illustrate the diversity of methods by the algorithmic richness of their experimental evaluation.
The central idea of the authors' method, ConfMOT, is to use the “confidence score” (CS) — the detector's measure of confidence — to control the Kalman filter and the object association process. The authors propose:

  • to adapt the measurement noise depending on the confidence score;
  • to include CS in the cost matrix used in the Hungarian algorithm;
  • to manage trajectory deletion based on accumulated confidence scores.
    Thus, we have a perfectly normal paper — a typical applied-level work in computer vision, with a clear idea and neat implementation.

Minor technical remarks

  1. The authors did not indicate ORCID, so it is difficult to assess their scholarly credentials. Populated scientific profiles are needed.
  2. The abstract is somewhat non-native. The reader must infer that the subject is video sequences, that the Kalman filter is used for sequential frame processing, etc. The style needs to be ordered.
  3. The Kalman filter algorithm is recorded very sloppily. Required: 1) explicitly state the motion model, 2) explicitly define all variables, 3) ensure consistency in the use of the time index t — for example, the derivative u lacks the index t (is it time-independent, a constant?).
  4. No variable in the text should remain without an explicit definition. For example, “c_t is the detection” is not mathematically well-posed, because there is no such notion as a “detection” in the formal sense, only a random function taking values, etc.
  5. The same applies to the variables in (5), (6), (7). I repeat: for model correctness one must specify not only the physical meaning of quantities but also define the mathematical object that represents them.
  6. Work on the English language is required: 1) several typos (“sine” instead of “since”, “shows” instead of “show”), 2) inconsistency in abbreviations (KF vs the KF, MOT vs the MOT), 3) some figure captions are too brief.
